# Cell-In-Cell Structures in Early Breast Cancer Are Prognostically Valuable

**DOI:** 10.3390/cells12010081

**Published:** 2022-12-24

**Authors:** Mareike F. Bauer, Laura S. Hildebrand, Marie-Charlotte Rosahl, Ramona Erber, Sören Schnellhardt, Maike Büttner-Herold, Florian Putz, Oliver J. Ott, Carolin C. Hack, Rainer Fietkau, Luitpold Distel

**Affiliations:** 1Department of Radiation Oncology, Universitätsklinikum Erlangen, Friedrich-Alexander-Universität Erlangen-Nürnberg (FAU), Universitätsstraße 27, D-91054 Erlangen, Germany; 2Comprehensive Cancer Center Erlangen-EMN (CCC ER-EMN), Universitätsklinikum Erlangen, Friedrich-Alexander-Universität Erlangen-Nürnberg (FAU), D-91054 Erlangen, Germany; 3Institute of Pathology, Universitätsklinikum Erlangen, Comprehensive Cancer Center Erlangen-EMN, Friedrich-Alexander-Universität Erlangen-Nürnberg (FAU), Krankenhausstraße 8–10, D-91054 Erlangen, Germany; 4Department of Nephropathology, Institute of Pathology, Universitätsklinikum Erlangen, Friedrich-Alexander-Universität Erlangen-Nürnberg (FAU), Krankenhausstraße 8–10, D-91054 Erlangen, Germany; 5Department of Gynecology, Universitätsklinikum Erlangen, Friedrich-Alexander-Universität Erlangen-Nürnberg (FAU), Universitätsstraße 21-23, D-91054 Erlangen, Germany

**Keywords:** cell-in-cell, breast cancer, non-professional phagocytosis, ionizing radiation, accelerated partial breast irradiation, radiotherapy

## Abstract

Cell-in-cell (CIC) structures in breast cancer have so far been studied in a small inhomogeneous patient population, suggesting the prognostic importance of CIC. In the present study, we focused on CIC in early hormone-sensitive breast cancer. With in vitro co-culture experiments, we compared the homotypic phagocytic capacity of two breast cancer cell lines to that of primary human fibroblasts. Afterward, we studied 601 tissue specimens from 147 patients participating in an institutional accelerated partial breast irradiation (APBI) phase II trial. Both breast cancer cell lines performed non-professional phagocytosis at a higher rate than primary human fibroblasts. In this study cohort, 93.2% of the patients had T1 tumours, and 6.8% had T2 tumours. CIC was found in 61.2% of the patients, with a CIC rate ranging from <1/mm^2^ to 556.5/mm^2^ with a mean of 30.9/mm^2^ ± 68.4/mm^2^. CIC structures were prognostically favourable for local recurrence-free survival and disease-free survival. Regarding metastasis-free survival, CIC-positive patients had an unfavourable prognosis. Subgroup analysis indicated a correlation between a high proliferation index and high CIC rates. CIC had the highest prognostic value in young breast cancer patients (*p* = 0.004). With this study, we provide further evidence of CIC as a prognostic marker in breast cancer.

## 1. Introduction

Cell-in-cell (CIC) structures arise from the engulfment of one cell by another non-phagocytic cell. They are found in a wide range of cancers and are therefore declared a “hallmark of cancer”, even though their biological impact is divergent [1]. By the engulfment of matrix-detached [2,3] or otherwise abnormal cells, cell-in-cell formations can prevent metastasis formation and malignant degeneration [4,5,6,7]. The tumour suppressive role of cell-in-cell structures can also be attributed to the release of cytotoxic molecules by engulfed immune cells inside host cells [8,9].

By contrast, CIC formation can have tumorigenic potential. Via cell competition, CIC formation is a selection mechanism with survival benefits for the most adaptable and malignant cell clones [1,4,10,11,12]. Furthermore, the engulfment of a neighbouring cell can lead to aneuploidy and thereby advance malignant transformation [13,14]. The engulfment of targeting immune cells can also be an immune evasion mechanism [3,15].

In our study, we focused on CIC in breast cancer. In 2020, breast cancer was the most frequently diagnosed cancer and caused the most cancer deaths in women [16]. Before considering treatment options, the heterogeneous group of breast cancer patients was divided into subgroups [17] depending on their hormone receptor status, Her2 receptor status, histological grade [18,19,20,21], stage [20,21,22], and proliferation index in Ki67 staining [18,21,23]. All these factors are important prognostic markers [18,19,20,21,22,23].

CIC has been reported as a potential prognostic marker in several tumours [14,24,25,26,27,28,29]. Zhang et al. studied CIC in a small cohort of breast cancer patients and identified CIC as a prognostic factor, with overall CIC as beneficial for patient survival [26]. In the study by Zhang et al., patients had mostly advanced cancer, with only 8.8% of T1 tumours [26]. We were interested in the prognostic value of CIC in a cohort of patients with early hormone receptor-positive breast cancer and the potential of CIC as a prognostic marker. The cohort was from a phase II accelerated partial breast irradiation (APBI) trial at the University Hospital Erlangen and was carefully pre-selected and homogeneous due to strict inclusion criteria [30].

## 2. Materials and Methods

### 2.1. In Vitro Experiments

To examine if breast cancer cells are capable of non-professional phagocytosis, we performed in vitro experiments with two breast cancer cell lines (MCF-7 and MDA-MB-231) and compared their phagocytic capacity to the capacity of a primary skin fibroblast cell line (SBLF-9). We performed each experiment three times and used different immunofluorescence staining.

According to the non-professional phagocytosis protocol established in our lab [31], we cultured half of the cells on round glass coverslips in a 5% CO_2_ atmosphere at 37 °C to obtain adherent cell layers. Media were prepared and comprised F-12 Medium, Dulbecco’s Modified Eagle’s Medium, foetal calf serum (FCS), and 1% penicillin/streptomycin antibiotics. The other half of the cells were stained with live dye CyTRAK orange (Thermo Fisher, Schwerte, Germany) and exposed to 56 °C hyperthermia in a water bath for 1 h. The red-stained cells were then co-incubated for 4 hours, with their homotypic cells grown as adherent cell layers.

For immunofluorescence staining, the cells were first permeabilized and fixed with 3.7% formaldehyde/0.1% Triton X-100 at room temperature. After washing three times with a phosphate-buffered saline (PBS), we incubated the cells overnight at 4 °C with a blocking solution comprising PBS with 5% FCS, 0.3% Triton X-100, and 0.3% sodium azide. The samples were washed three times with PBS and incubated with the primary antibody (α-Tubulin, rabbit, 1:250, cat# ab52866, abcam, Cambridge, UK) overnight at 4 °C in a humidity chamber. The next day, three washing steps with PBS preceded 1.5 h of incubation with the secondary antibody (goat anti-rabbit IgG Alexa Fluor 488, cat# A-11008, Thermo Fisher, Schwerte, Germany) in a humidity chamber at room temperature. The excess antibody was removed by three washing steps with PBS before drying the samples and mounting them with Prolong Gold and DAPI (4′,6-diamidino-2-phenylindole) (Thermo Fisher, Schwerte, Germany). Primary and secondary antibodies were diluted in PBS containing 0.1 g bovine serum albumin and 30 µl Triton X-100. CIC rates were analysed semi-automatically using Biomas image analysis software. Blue nuclei were automatically labelled, and confluent nuclei were separated by area size and shape. Next, the area of the green cells and the centre of the red cells were labelled. If a red cell label was inside a green area, it was marked as CIC. In all steps, it was possible to add or delete cells manually by the user. Prior to transferring the CIC rates to a spreadsheet, all CIC structures were displayed again and had to be approved by the user.2.2. Tissue microarray (TMA) analysis

A cohort of 147 patients was treated for early-stage breast cancer at the University Hospital Erlangen between 2000 and 2005 as part of a prospective APBI phase II trial. APBI was performed with interstitial multicatheter brachytherapy. The inclusion criteria for this study were the histopathologically confirmed invasive breast carcinoma of any histology with a diameter of ≤3 cm, unifocal and unicentric breast cancer, and clear resection margins of at least 2 mm in any direction. Other criteria were hormone sensitivity (ER+/PR+, ER+/PR-, ER-/PR+), histological grade 1 or 2, no lymphatic or blood vessel invasion, pN0/pNmi, and no distant metastases [30,32]. All patients received breast-conserving surgery and brachytherapy [33,34,35]. Clinical data and pathological features were prospectively collected in the APBI study (Table 1). 

For this study, follow-up time was restricted to 120 months. The TNM stage ranged from T1mic to T2. A total of 601 paraffin-embedded samples from the tumour centre, tumour infiltration zone, healthy tissue proximal and distal to the tumour, lymph nodes, and lymph node metastases were transformed into 12 tissue microarrays and were immunohistochemically stained for E-cadherin (anti-CD324, clone 36/E-Cadherin, cat# 610182, 1:2000, BD Biosciences, Heidelberg, Germany). Visualization was performed on a Ventana BenchMark ULTRA stainer (Roche) using an UltraView DAB IHC Detection Kit (Roche) for detection. Hematoxylin-eosin staining was used for nuclear counter-staining.

Written informed consent was obtained “front door” from all patients for the collection of their tissue and clinical data. The use of formalin-fixed paraffin-embedded material from the Archive of the Institute of Pathology was approved by the Ethics Committee of the Friedrich-Alexander-University of Erlangen-Nuremberg on 24 January 2005 (21_ 19 B), waiving the need for consent to use the existing archived material.

The AxioImager Z2 (Zeiss, Göttingen, Germany), a fluorescence microscope, was used to acquire images for both cell culture experiments and TMAs. They were analysed with Biomas software. Initially, the area of the stained tissue spot was automatically marked, and the area of the epithelium was calculated. Whenever a cell was completely engulfed by another cell and deformed the recipient cell’s nucleus, the resulting structure was classified as a CIC structure and was flagged manually by a mouse click. The ratio of CIC structures to recipient cells is expressed as the CIC rate. Biomas then automatically calculated the CIC per mm^2^ in the tissue microarrays and transferred it to a spreadsheet [36].

### 2.2. Statistical Analysis

We used GraphPad Prism Version 8 for data analysis and plotting. For survival analyses, Kaplan–Meier curves were calculated using IBM SPSS statistics software. To determine differences, Log-rank tests were performed. Kruskal–Wallis tests were performed to determine the differences between the cell lines in the in vitro experiments. Correlation analyses were also performed using IBM SPSS statistics software. The Cox proportional hazards model was used to calculate the hazard ratios of CIC rates in central cancer and clinicopathological characteristics. Covariates with *p* < 0.3 in univariate analysis were included in multivariate Cox regression. The proportional hazards assumption was tested by the visual inspection of log minus log curves and was found to be satisfactory for all multivariate covariates. *p*-values < 0.05 were considered statistically significant.

## 3. Results

### 3.1. Phagocytic Capacity of Breast Cancer Cell Lines

As proof of principle, we studied the phagocytic capacity of breast cancer cells in vitro. Both studied breast cancer cell lines (MCF-7, MDA-MB-231) performed homotypic non-professional phagocytosis. The host cells completely engulfed the red-stained target cells, thereby forming CIC structures (Figure 1A,B,D,E). The CIC rates of MCF-7 and MDA-MB-231 were higher than the CIC rate of primary human fibroblasts (SBLF-9) (Figure 1C,F) with 1.5% ± 0.6%, 1.9% ± 1.4% and 0.6% ± 0.4%, respectively (Figure 1G). However, the differences did not vary significantly with *p*-values of 0.539 in the Kruskal–Wallis tests. 

### 3.2. CIC in Tissue Microarrays (TMAs) 

We studied 601 tissue spots on TMAs from 147 breast cancer patients with early hormone receptor-positive tumours. The TMAs were immunohistochemically stained for E-cadherin, and the CIC structures were counted. The stained structures inside a cell are the E-cadherin-stained membrane of the phagocytosed cell inside the host cell (Figure 2).

The TMAs originated from tumour tissue, healthy tissue, and lymph nodes at similar frequencies, and a small number originated from metastatic tissues. (Figure 3A). Up to six samples per patient were included. Overall, we found CIC in 21.8% of the tissue spots. The highest proportion of cases with CICs was found in the central tumour sections (51.6%) and the tumour infiltration zone (41.2%) (Figure 3B). The CIC rate per mm^2^ ranged from < 1/mm^2^ to 556.5/mm^2^ for CIC-positive tissues with a mean of 30.9/mm^2^ ± standard deviation (SD) 68.4/mm^2^ and a mean of 6.8/mm^2^ ± SD 34.3/mm^2^ for all tissues (Figure 3C). The highest rates were found in the infiltration zone (14.3/mm^2^ ± SD 27.8/mm^2^) and central tumour tissue (7.6/mm^2^ ± SD 11.4/mm^2^). CIC rates were similar in proximal (8.1/mm^2^ ± SD 17.2/mm^2^) normal tissue and were lower in the distal (4.6/mm^2^ ± SD 21.9/mm^2^) normal tissue and lymph nodes (1.2/mm^2^ ± SD 8.2/mm^2^) (Figure 3D).

In this cohort, only patients with TNM stages of up to pT2 (≤3 cm) were included. A total of 83.0% of the samples originated from patients who staged pT1b or pT1c. Only 6.8% of patients had pT2 cancer (Table 1). Throughout the T stages, the percentage of CIC-positive TMAs was comparable except for pT1a (Figure 3E). Here, only 22.2% of the analysed tissue spots contained CIC structures. Most patients included in this hormone receptor-positive study cohort had luminal A-like tumours. Only 5.4% had luminal B-like Her2-positive tumours (Table 1). CIC was found in 75.0% of the tissue sections from Her2-positive cancers. In Her2-negative cancers, non-professional phagocytosis was found in slightly fewer TMAs (Figure 3F).

While we found CIC structures in at least one TMA of 61.2% of the patients, adverse events such as metastasis formation, tumour recurrence, or death were rare during the follow-up period of 10 years (Figure 4A). The CIC counted in the central cancer region had the highest association with survival. Patients with CIC in the central cancer tissue clearly had a favourable prognosis regarding local recurrence-free survival (Figure 4B) (*p* = 0.008) and disease-free survival (Figure 4C) (*p* = 0.027). Multivariate analyses revealed that Ki67 (*p* = 0.003) and CIC in central cancer (*p* = 0.018) were independent risk factors for disease-free survival (Table 2). In contrast to this favourable prognostic feature, CIC did not have a similar impact on the metastasis-free survival of the whole study cohort (Figure 4D) (*p* = 0.498). 

The CIC in the invasive front was not prognostically relevant, either for recurrence-free survival in the breast or for disease-free survival or metastasis-free survival (Figure 5A–C) (*p* > 0.237). The combination of the CIC rates of central cancer and the invasive front also proved insignificant for survival (Figure 5D–F) (*p* > 0.142). Here, there was a slight tendency for high CIC rates to favour survival in in-breast recurrence-free survival and disease-free survival. In metastasis-free survival, however, high CIC rates tended to be unfavourable.

To know which group had the favourable prognosis, those without CIC in the central tumour and in the invasion front were compared with those who had CIC either only in the central tumour or only in the invasion front and those who had CIC in both the central tumour and the invasion front. Finding CIC both in central cancer and invasive front tissue spots was beneficial for local recurrence-free survival and disease-free survival, whereas an absence of CIC was unfavourable (Figure 5G,H). The tendency that the absence of CIC leads to a favourable prognosis for metastasis-free survival (Figure 5C,F) (*p* = 0.237, *p* = 0.375) is especially remarkable. Regarding metastasis-free survival, CIC in central cancer and the invasive front is most unfavourable (Figure 5J). 

After the overall analysis of our data, we studied the role of CIC in central tumour tissue spots for the disease-free survival of different subgroups. We divided the T-stages into two subgroups, one comprising T-stages pTmic, pT1a, and pT1b and the other comprising pT1c and pT2. There were no differences in the mean CIC rates (Figure 6A) (*p* = 0.338), and the prognostic values were similar in both groups (Figure 6B,C) (*p* = 0.122 vs. *p* = 0.120). The proliferation marker Ki67 was split at 10%. Distinctly higher CIC rates occurred in the high proliferation group of Ki67 > 10% (Figure 6D). However, CIC had a higher prognostic relevance in the low proliferation group (Figure 6E,F) (*p* = 0.037 vs. *p* = 0.155). Age did not have an influence on the CIC rates (Figure 6G). However, the prognostic value of CIC was high at an age of 60 years or younger (Figure 6H) (*p* = 0.004), while at older ages, CIC rates were not prognostically relevant (Figure 6I) (*p* = 0.784).

## 4. Discussion

With this study on 147 patients, we provide further insights into the prognostic significance of CIC in breast cancer. Consistent with the findings of Zhang et al. [26], the presence of CIC structures in stage T1 tumours was beneficial for the patients’ local recurrence-free and disease-free survival, suggesting the protective role of CIC formation in early breast cancer. These findings correspond to theories of a tumour suppressive function in CIC formation [12], amongst others, by eliminating the matrix detached cells [2,3], preventing genetic disruption and malignant transformation [5], and thereby maintaining tissue homeostasis [4,6,7].

In many cancers, CIC is an indicator of a higher metastatic potential [10,11,12,13,14,27,28,29,37,38]. While there was no significant difference in metastasis-free survival in patients with or without CIC in their TMAs, the presence of CIC in the central tumour and tumour infiltration zone tended towards an unfavourable outcome. Our findings, therefore, point to the opposing effects that CIC formation can have in tumour biology [12]. The divergence might be explained by focusing on the mechanism behind CIC formation [12] or the differences in host and target cells, creating a heterologous population of CIC structures [26,29]. Thus, in-depth studies on CIC subtypes in a bigger cohort of breast cancer patients, as suggested by Zhang et al. [26], are necessary to determine the prognostic value of CIC in breast cancer. 

A limitation of our study is the relatively low number of adverse events during follow-up due to effective treatment options, which led to high survival rates in the early stages of breast cancer. This limits the significance of our study regarding CIC as a prognostic marker. Future studies should focus on breast cancers larger than T2 and Her2 positive and triple-negative breast cancer, as these subgroups are hitherto underrepresented in existing studies. Moreover, CIC should be considered as an additional marker to the well-established ones, as the prognostic impact differs with tumour stages and the molecular characteristics of the tumour tissue.

## 5. Conclusions

Our findings indicate that CIC structures are potential prognostic biomarkers with both beneficial and adverse impacts depending on the breast cancer subtype and probably the biology of CIC formation. CIC analysis might contribute to a more personalised and precise therapeutic approach by the further subdivision of the commonly known molecular subtypes of breast cancer into more refined prognostic categories.

## Figures and Tables

**Figure 1 cells-12-00081-f001:**
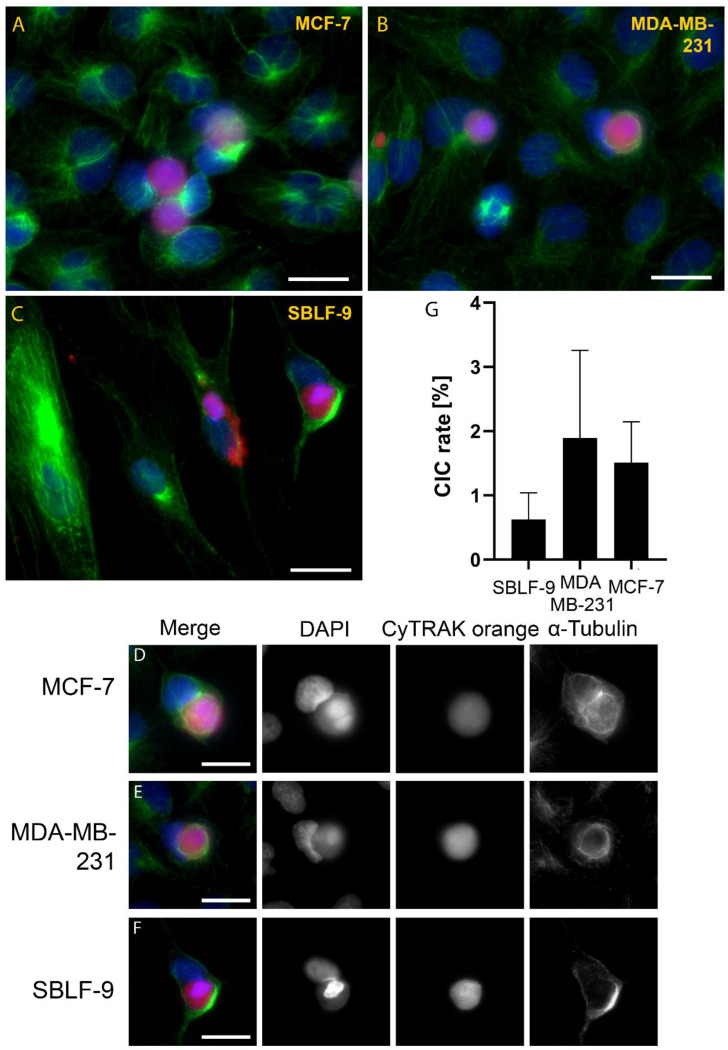
Non-professional phagocytosis assay in vitro. Overview images from live-stained, hyperthermia-treated target cells (red) were added to an adherent layer of host cells (α-Tubulin, green) to assess the homotypic phagocytic capacity of two breast cancer cell lines (**A**) MCF-7, (**B**) MDA-MB-231, and (**C**) a human primary fibroblast cell line. Nuclei are stained with DAPI (blue). Single (**D**) MCF-7, (**E**) MDA-MB-231, and (**F**) human primary fibroblast cells after phagocytosis visualised in single colours with the nuclei in blue (DAPI), and the dead cells (CyTRAK orange) and α-tubulin in green. The images displayed are representative for >150 CIC structures analysed. (**G**) Comparison of means of the cell-in-cell rate between the three cell lines studied.

**Figure 2 cells-12-00081-f002:**
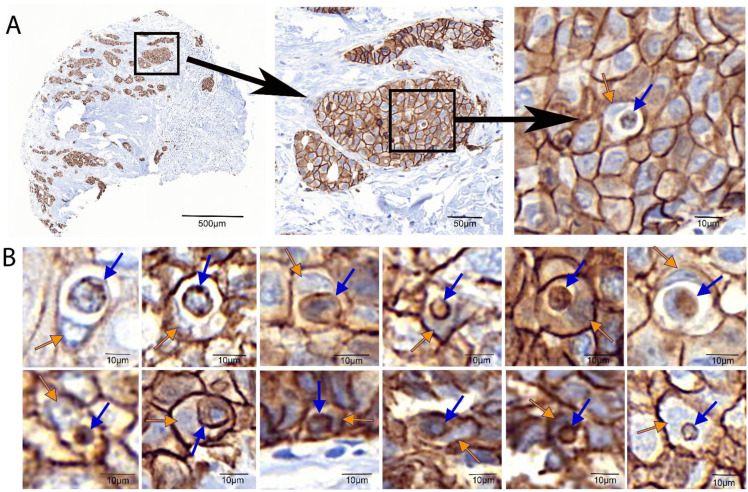
Analysis of tissue microarrays (TMAs) from breast cancer patients. TMAs were stained for E-cadherin to visualize cell membranes. (**A**) Individual TMA spots were manually screened for cell-in-cell (CIC) structures, which were flagged in the images. After manual analysis, Biomas software counted and flagged CIC structures and tumour cells to calculate CIC rates and CIC per mm^2^. (**B**) Examples of CIC found throughout the TMA analysis. Blue arrows indicate the E-cadherin cell membrane of the phagocytosed cell inside the host cell. Orange arrows indicate the nucleus being pushed to the side in the host cell.

**Figure 3 cells-12-00081-f003:**
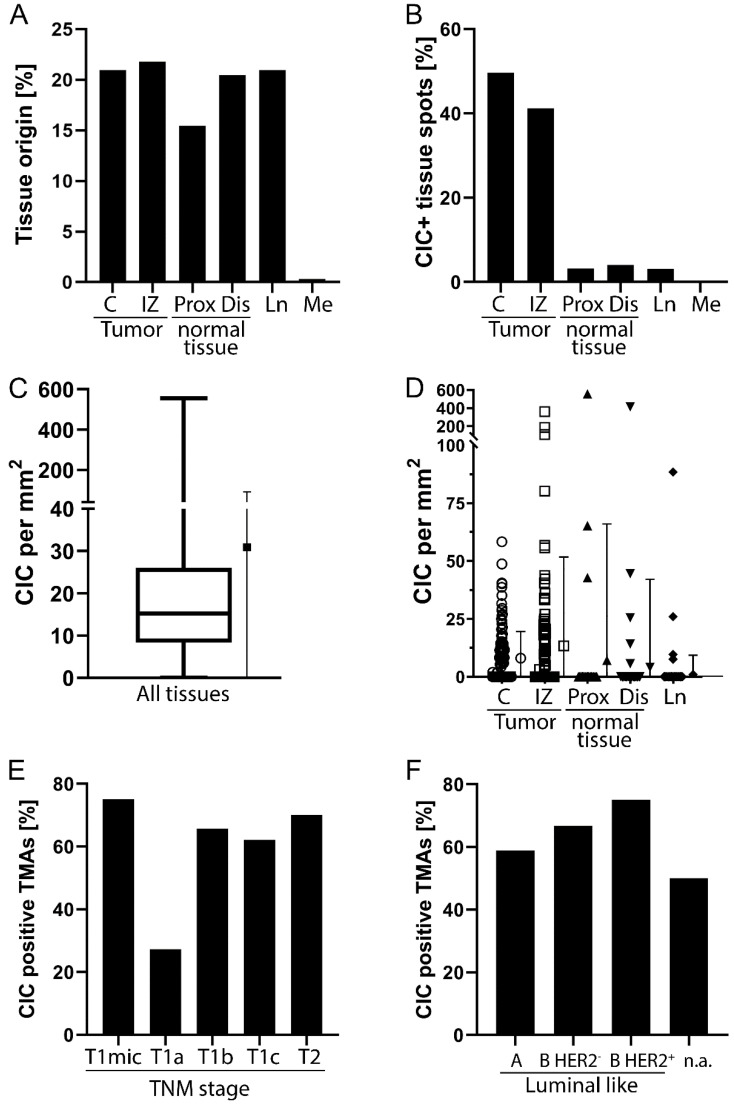
Statistical analysis: (**A**) Tissue origin and (**B**) Respective proportion of cell-in-cell (CIC) positive tissue spots. (**C**) Overall CIC per mm^2^. (**D**) CIC per mm^2^ in the different compartments. (**E**) Proportions of CIC positive tissue spots depending on TNM stage of analysed tumours. (**F**) CIC positive tissue spots depending on molecular subtype of cancers. C = central, IZ = infiltration zone, Prox = proximal, Dis = distal, Ln = lymph node, Me = metastasis, T = tumour size from T-category of the TNM stage, n.a. = not applicable. The TNM stage and luminal category were determined accordingly to the guidelines [17].

**Figure 4 cells-12-00081-f004:**
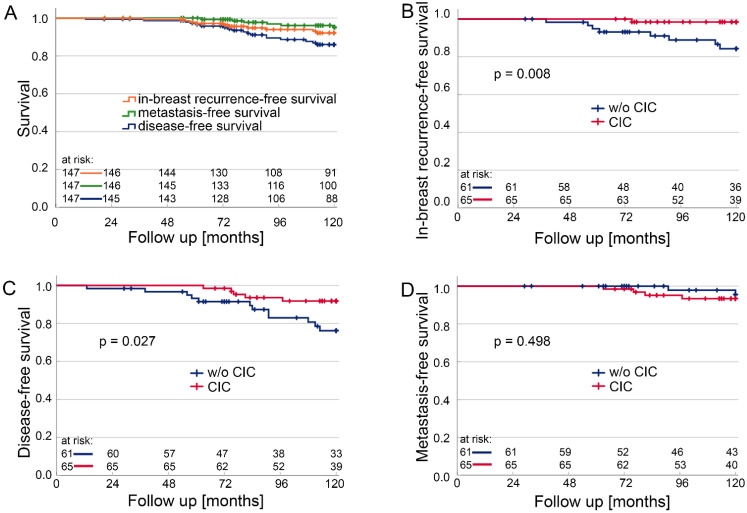
Kaplan–Meier plots comparing patients with tissue without cell-in-cell structures (CICs) and patients with tissue and CICs. (**A**) Local recurrence free survival, metastasis-free survival, and disease-free survival of the total cohort. (**B**) Local recurrence-free survival, (**C**) Disease-free survival, and (**D**) Metastasis-free survival.

**Figure 5 cells-12-00081-f005:**
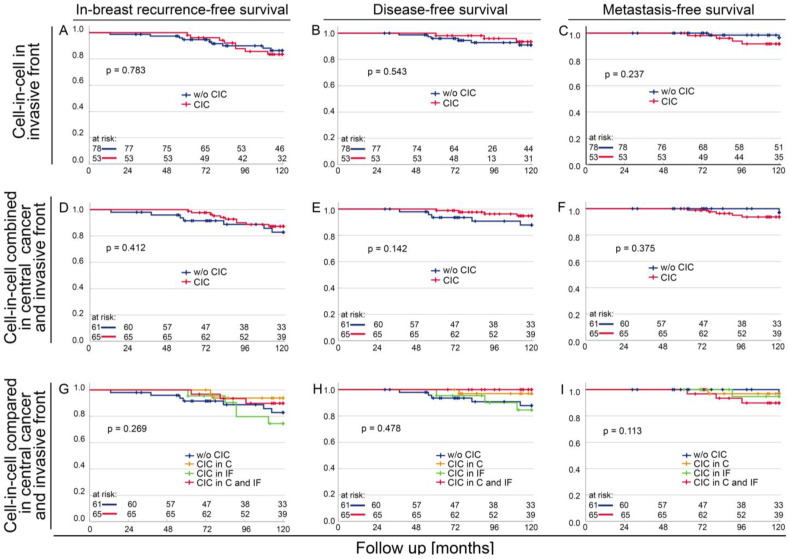
Kaplan–Meier plots depending on cell-in-cell (CIC) origin. Patients with tissue without CICs and patients with tissue and CICs were compared and Kaplan–Meier plots were calculated for (**A**) Local recurrence free survival, (**B**) Disease-free survival, and (**C**) Metastasis-free survival. CIC combined in central cancer and invasive front were determined and Kaplan–Meier plots were calculated for (**D**) Local recurrence free survival, (**E**) Disease-free survival, and (**F**) Metastasis-free survival. CIC in central cancer and invasive front or central cancer or invasive front without CIC were compared and Kaplan–Meier plots were calculated for (**G**) Local recurrence free survival, (**H**) Disease-free survival, and (**I**) Metastasis-free survival of the total cohort.

**Figure 6 cells-12-00081-f006:**
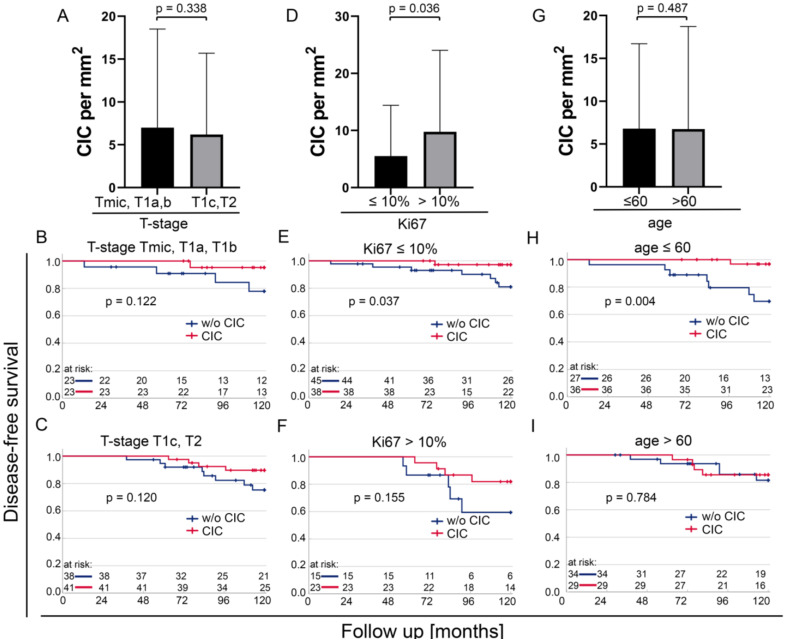
Prognostic value of cell-in-cell (CIC) in different subgroups. (**A**) CIC per mm^2^ in the group comprising pTmic, pT1a, and pT1b compared with the group comprising pT1c and pT2 and the importance of CIC rates in the (**B**) pTmic, pT1a, pT1b subgroup and (**C**) pT1c, pT2 subgroup. (**D**) CIC per mm^2^ in tissues with Ki67 ≤ 10% compared with Ki67 > 10% and the prognostic significance of CIC rates in the subgroups of (**E**) Ki67 ≤ 10% and (**F**) Ki67 > 10%. (**G**) CIC per mm^2^ in patients 60 years of age or younger compared with patients older than 60 years, and the prognostic significance of CIC rates in the subgroup (**H**) Age ≤ 60 years and the subgroup (**I**) Older than 60 years.

**Table 1 cells-12-00081-t001:** Characteristics of the study cohorts.

Variables		Early Hormone Receptor Positive (%)
T stage	T1mic	4 (2.7)
	T1a	11 (7.5)
	T1b	35 (23.8)
	T1c	87 (59.2)
	T2	10 (6.8)
N stage	N0	14 (98.0)
	N1	3 (2.0)
M stage	M0	136 (92.5)
	M1	11 (7.5)
Grade	G1	40 (27.2)
	G2	102 (70.1)
	G3	4 (2.7)
Ki67	<10%	99 (67.3)
	≥10%	48 (32.7)
Oestrogen receptor status	ER-positive	143 (97.3)
	ER-negative	4 (2.7)
Progesterone receptor status	PR-positive	134 (91.2)
	PR-negative	13 (8.8)
HER2	HER2-	9 (6.1)
	HER2+	135 (91.8)
	n.a.	3 (2.0)
Molecular subtype	Luminal A-like	90 (61.2)
	Luminal B-like (Her2 negative)	39 (26.5)
	Luminal B-like (Her2 positive)	8 (5.4)
	n.a.	10 (6.8)

**Table 2 cells-12-00081-t002:** Univariate and multivariate analysis of disease-free survival according to Cox’s proportional hazards model.

Breast Cancer	Univariate Analysis			Multivariate Analysis		
Variable	Hazard Ratio	95% C.I.	*p*	Hazard Ratio	95% C.I.	*p*
Age in years (younger 60 years [*n* = 51] v. older 60 years [*n* = 51 ])	0.435	0.104-1.827	0.256	0.596	0.167-2.124	0.425
T category (Tmic. T1a. b [*n* = 31] v. T1c. T2 [*n* = 71 ])	1.689	0.36-7.928	0.507	---	---	---
Grad (1 [*n* = 73] v. 2. 3 [*n* = 29 ])	2.514	0.428-14.759	0.307	3.002	0.592-15.229	0.185
Stage (UICC I [*n* = 36] v. UICC II and higher [*n* = 66 ])	2.452	0.182-32.996	0.499	---	---	---
Ki67 (≤10% [*n* = 92] v. >10% [*n* = 10 ])	8.387	2.015-34.902	0.003	7.016	1.941-25.366	0.003
Tumour size (mm) (≤15 [*n* = 68] v. >15 [*n* = 34 ])	0.075	0.006-0.968	0.047	0.147	0.018-1.178	0.071
Her2 status (negative [*n* = 95] v. positive [*n* = 7 ])	2.611	0.219-31.129	0.448	---	---	---
ER status (negative [*n* = 2] v. positive [*n* = 100 ])	117.184	0-251.235	0.989	---	---	---
PR status (negative [*n* = 9] v. positive [*n* = 93 ])	0.353	0.035-3.604	0.380	---	---	---
CIC in central cancer (w/o CIC [*n* = 51] v. CIC [*n* = 51 ])	0.134	0.026-0.692	0.016	0.192	0.048-0.757	0.018

## Data Availability

The datasets used and analyzed during the current study are available from the corresponding author on reasonable request.

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
