# Peer review of "Cell-In-Cell Structures in Early Breast Cancer Are Prognostically Valuable"

_cells, 2022, doi:10.3390/cells12010081_

Round 1
Reviewer 1 Report
This excellent manuscript studied cell-in-cell (CIC) structures in breast cancer. The authors performed in vitro co-culture experiments and then validated their findings in a large, well-organized patient cohort. I am confident with their results and believe they are essential to the breast cancer field. Hence I suggest acceptance of the manuscript in its current form.
Author Response
We thank you very much for this very positive assessment of our article and are very pleased about it.
Reviewer 2 Report
In this manuscript, authors presented research regarding CIC structures with
respect to their prognostic importance in breast cancer. However, the study
content find a lack of innovation and novelty due to already published
information on the related content.
Author Response
We are somewhat disappointed by this assessment, as we do contribute something new to the scientific discourse. There have been no studies of such early stages of breast cancer and certainly no comparable homogeneous cohort studied for CIC. We were able to show that CIC are already present in large quantities in such early tumours and are also significant for the prognosis. We see this as an important finding in order to be able to assess the significance of CIC as a whole in the future.
Reviewer 3 Report
The paper by Bauer et al. describes CIC structures as a potential prognostic biomarker with beneficial and adverse impact depending on the status or type of breast cancer using a study of samples from 147 patients. Although the significance of this study is mainly focused in early stages of breast cancer and narrows in subsequent advanced tumour stages due to the effectiveness of treatments applied, this analysis sheds light on CIC as a prognostic marker, at least in some sub-types of breast cancers.
The paper is well written and the analysis of the data are well treated and described. I have only minor suggestions to improve the manuscript:
1. Could you specify the antibodies used in this study with reference number and concentrations used for immunofluorescence staining?
2. Could be possible to remake the Figure 1 showing a bigger area of cells from the three different cell lines? This would bring the possibility of flagging positive and negative cells for phagocytosis and how looks, also pointing in the figure legend. I think it would be useful for readers, as it’s not completely clear.
3. Figures 3, 4, 5 and 6 (B-J) are difficult to read as all the graphs and texts are really close each other. Could you use some space between numbers, titles and different panels making the figure easier to follow?
4. The proliferation marker Ki67 was split at 10% for analysis (Figure 6D, E, F). Have you tried to increase the number of groups to maximize the differences?
5. Same as 4 for Figure 6H, splitting the category <60 years into different groups.
6. Figure 5 should be organized as cited in the text, sometimes is hard to follow.
7. I miss more description from Figure 5 in the text. It would clear the results presented.
All in all, I overall support publication.
Reviewer 4 Report
The manuscript is about cell-in-cell (CIC) structures which result from engulfment of one cell by another.
The authors focussed on the prognostic value of CIC in a cohort of patients with early hormone receptor-positive breast cancer and the potential of CIC as a prognostic marker.
Comments
Methods
1. Page 2, lines 78-88. Please mention which primary and secondary antibodies were used.
2. Could you mention a reference for the recognition of the CIC structure to lines 112-115?
Results
Figures 1 and 2.
1. The two figures try to show the proof of principle and the implamantation on the TMA. It would be benefitial if the CIC could be presented on the cell lines with the same markers or comparable technique used as presented in Figure 2 in tissue.
2. Figure 2 requires better description. "A" is DAB detected "B" shows a red stain. Please describe what is brown and what is red, what is the counterstain. Please use scale bars on the Figures 1 and 2. Please explain what are these structures where the arrows point to. Are these complete E-cadherin-labelled structures inside other cells?
3. If more staining techniques used for detecting the CIC structures than were they equally relevant for the analysis? Is the red method more sensitive?
4. Could you give some hints on how the Biomas software counted the flagged CIC structures and how they were flagged?
5. Figure 3. Not all components on the Figure are explained. For example. What is "T" on panel C?
Round 2
Reviewer 2 Report
Though CIC contributes its importance with respect to breast cancer study, however the manuscript lacks its uniqueness, as, similar studies are already reported whose references are attached follows: 1. Zhang, X., Niu, Z., Qin, H., Fan, J., Wang, M., Zhang, B., ... & Sun, Q. (2019). Subtype-based prognostic analysis of cell-in-cell structures in early breast cancer. Frontiers in oncology, 9, 895. 2. Mousavi, S. S., & Razi, S. (2021). Cell-in-cell structures are involved in the competition between cells in breast cancer. arXiv preprint arXiv:2112.13271. 3. Wang, M., Ning, X., Chen, A., Huang, H., Ni, C., Zhou, C., ... & Sun, Q. (2015). Impaired formation of homotypic cell-in-cell structures in human tumor cells lacking alpha-catenin expression. Scientific reports, 5(1), 1-9.Author Response
Dear reviewer,
Thank you for the constructive criticism. We edited our manuscript according to your recommendations. We hope our manuscript is now appropriate for publication in “Cells”.
Thank you very much for pointing out these interesting works about CIC in breast cancer.
Zhang et al studied a wider range of different breast cancer patients. Only a small amount were early hormone receptor positive. As we focused on early hormone receptor positive breast cancer patients, we provide additional data on early breast cancer to the study group analyzed by Zhang et al. Throughout our work, we put our findings in conjunction with their work.
We added a reference to Wang et al.’s study to our manuscript as it provides valuable insights into the mechanism of CIC formation of various breast cancer cell lines.
Reviewer 4 Report
All comments are answered sufficiently.
Author Response
Dear reviewer,
thank you very much for your positive assessment of our article.